# Effects of Nordic Walking Training on Anthropometric, Body Composition and Functional Parameters in the Middle-Aged Population

**DOI:** 10.3390/ijerph19127433

**Published:** 2022-06-17

**Authors:** Alessia Grigoletto, Mario Mauro, Alberto Oppio, Gianpiero Greco, Francesco Fischetti, Stefania Cataldi, Stefania Toselli

**Affiliations:** 1Department of Biomedical and Neuromotor Sciences, University of Bologna, 40126 Bologna, Italy; alessia.grigoletto2@unibo.it (A.G.); stefania.toselli@unibo.it (S.T.); 2Department of Life Quality Studies, University of Bologna, 47921 Rimini, Italy; mario.mauro4@unibo.it; 3School of Pharmacy, Biotechnology and sport Science, University of Bologna, 40126 Bologna, Italy; alberto.oppio@studio.unibo.it; 4Department of Basic Medical Sciences, Neuroscience and Sense Organs, University of Study of Bari, 70124 Bari, Italy; stefania.cataldi@uniba.it

**Keywords:** anthropometric measure, body composition, middle-aged, Nordic walking, outdoor physical exercise, physical exercise

## Abstract

Nordic walking (NW) is an easy physical exercise that is usually proposed for clinical populations and for the elderly. The aim of the present study was to examine the effects of a period of NW training in a non-clinical middle-aged population on anthropometric, body composition and functional parameters. A pre-test/post-test study design was conducted on 77 participants: 56 women (72.7%, age 55.53 ± 9.73 years) and 21 men (27.3%, age 60.51 ± 8.15 years). The measurements were carried out with physical tests at the baseline and at the follow up. Participants did two weekly NW training sessions of about 60 min each. A questionnaire was administered to evaluate their feelings after the training period. Paired Students’ test was carried out to evaluate the pre–post differences, and the analysis of variance was performed to evaluate the questionnaire. Participants had significantly less stress and anxiety after the NW training. Body fat parameters showed a significant decrease, especially for women. Phase angle and strength of lower body presented a significant increase in both sexes after the training period. In conclusion, NW shows many potential benefits also for the nonclinical population and could be an important exercise to remain active and to maintain a good health condition.

## 1. Introduction

The importance of physical activity is well-established, even if it is estimated that 3.3 million people die annually worldwide due to physical inactivity [1]. Physical inactivity is one of the five leading risks for mortality globally because it is responsible for increasing the risk of chronic disease and cancer [2]. So, it is important to identify forms of physical activity that are easily accessible and that can be performed by a large number of people to improve fitness and health status [3]. This aspect is important for people of all ages, especially for the middle-aged and elderly. To improve public health, green space represents an ideal environment which promotes PA due to its easy accessibility and no cost [4,5]. Additionally, a recent review showed that participants who practiced PA with a contact with nature improved their psychological well-being [6]. In addition, some studies evidenced that long-term adherence to exercise initiatives conducted in an outdoor natural environment or urban green space is superior to that of indoor exercise interventions [7]. Despite these studies, green spaces are generally underutilized, and visitors are often engaged in low levels of physical activity during their visits [6,8,9].

Nordic walking (NW) is a particular kind of walking in which specially designed poles are used [10]. NW was initially designed for cross-country skiing athlete’s training during the summer [11], and since then it has gained popularity worldwide as a health-promoting activity and as a physical exercise [7]. By incorporating upper body muscle activity similar to that in cross-country skiing, NW incorporates a total body version of walking with a greater caloric expenditure due to the higher amount of muscle mass used and potentially enhanced physical fitness benefits [11,12]. According to previous systematic reviews, NW has a lot of important benefits for people [3,13], such as resting the heart rate, blood pressure, exercise capacity, maximal oxygen consumption and quality of life. For these reasons, NW is suitable for prevention, and nowadays, it is more and more frequently recommended for the elderly [14]. Fritschi et al., in their review, found that most of the NW practitioners were mid to older aged men and women, from the clinical population (i.e., a diagnostic medical condition) [13]. The population studies included people suffering from diabetes, cardiovascular disease, peripheral artery disease, muscle-skeletal conditions, chronic obstructive pulmonary disease, Parkinson’s disease and breast cancer [15,16,17,18,19,20,21,22,23]. There were two studies in the nonclinical population, but both participants were only elderly women [11,24]. The effects of NW on the nonclinical population have been little investigated. 

In the middle-aged and elderly population, one of the main problems to counteract is the decline in lean mass and the increase in fat mass, in addition to the decline in strength. Body composition is an important index of health and nutritional status, aging and functional capacity [25]. In fact, the decline of body composition increases the risk of age-related diseases [26]. So, monitoring body composition has become crucial to evaluating the nutritional status in elderly and middle-aged people. Reference methods for assessing body composition present a high accuracy but are time-consuming and costly, and so cannot easily be applied on a regular basis [27]. Bioelectrical impedance analysis (BIA) can be considered practicable given that it is easy to use; in addition, precision and accuracy is usually reasonably high, with the latter being within 3.5 to 5% [28,29,30]. In recent years, one of the most popular methods to evaluate body composition was the bioelectrical impedance analysis (BIA) due to a combination of cost-efficiency, user friendliness and portability [31,32]. Several studies compared BIA with other reference methods and concluded that BIA has an accepted validity for the body composition assessment, especially in the evaluation of body fluids, fat free mass and fat mass [27,33,34,35,36]. By employing bioimpedance-based predictive equations, it is possible to estimate and monitor changes in body composition parameters such as fat mass, total body water and muscle mass [10]. In addition, using the qualitative analysis, it is possible to estimate the body composition through the raw bioimpedance parameters (resistance (R) and reactance (Xc)) as a point on the R-Xc graph in which both length and slope are considered. The vector slope indicates the integrity of the cell membrane and extracellular/intracellular (ECW/ICW) ratio [26,37]. Previous studies have shown that the PhA can be modulated by exercise. These studies showed significant changes in PhA after a resistance training program with a frequency of two or three times [38,39,40]. Tomeleri et al. (2018) and Souza et al. (2016) investigated the effects of resistance training on PhA in older women. Both studies found an inverse relation between PhA and inflammatory biomarkers [38], and a positive association with cellular health [40]. The same kind of result was found by Ribeiro et al. (2017) in young adults: an increase in PhA and a rise in cellular hydration after the period of training [39].

To our knowledge, there are no studies which consider NW and body composition together in healthy middle-aged and elderly population. The aim of the present study was to value the effectiveness of a period of NW training in counteracting the aging effects on body composition, BIVA patterns and functional parameter measurements (resistance, handgrip and lower body strength) in a healthy middle-aged population. In addition, a valuation regarding the attitude toward the green urban space and the appreciation in doing outdoor PA was performed.

## 2. Materials and Methods

### 2.1. Study Design and Participants 

This is a longitudinal study design with 3 months of follow-up and two measurements, one at baseline and one after the training treatment. Recruitment occurred thanks to the sport society “Nordic walking in Italy”, specifically with the headquarters of Venice. They do activities throughout the province of Venice, in the city parks, along the banks and always in the open air. They manage different walking groups in Mestre, Marghera, Spinea and Martellago. The study protocol was explained to the 94 members of the sport society, and those who voluntarily decided to participate in the study were included. In total, 19% of the members of the sport society decided to not participate, and 6% did not complete the entire period of training. Participants had to meet the following criteria to qualify for inclusions: (1) not have a chronic disabling disease, (2) not be bedridden, institutionalized or hospitalized, (3) be independently mobile without requiring human assistance or the aid of devices such as crutches, walkers, etc., (4) be without amputations, and (5) not have a pacemaker or the presence of chronic metabolic diseases. All participants signed an informed consent to participate in the study.

The study was approved by the Bioethics Committee of the University of Bologna (prot. N. 022254).

### 2.2. Intervention Training Programs

The baseline was set at the end of February 2021 after the stop related to the pandemic situation in Venice due to COVID-19, while the post-test was done in June 2021. Participants did two weekly training sessions of about 60 min each. Every training session included a 10 min warm-up, a 45 min main part during which people marched in the park following their trainer and a final 5 min of relaxing and stretching exercises. Three instructors followed the groups in different parks, and they were instructed to propose the same kind of training to the different groups, with the same kind of intensity. The intensity of training was decided a priori with the rate of perceived exertion, the Borg scale [41], and it was set at 5 on a scale of 10 points. Five means that it was a moderate activity and participants were able to talk and hold short conversation. 

### 2.3. Anthropometric Characteristics

The anthropometric measures were recorded at baseline and after the training period. Each participant’s height was recorded to the nearest 0.1 cm with a standing stadiometer (GPM, Steckborn, Switzerland), and body mass was measured to the nearest 0.1 kg using calibrated electronic scales (Seca, Basel, Switzerland). Body mass index (BMI) was calculated as the ratio of body weight to height squared (kg/m^2^), and the WHO cut-off was used to estimate the weight status of the subjects; less than 18.5 was classified as underweight, from 18.5 to 24.9 was considered normal weight, from 25 to 29.9 was overweight and more than 30 was classified as obese [42]. The operator took the following circumferences: relaxed arm, contracted arm, waist, hip and calf. All the circumferences were taken to the nearest 0.1 cm using a non-stretchable tape measure (GPM, Steckborn, Switzerland). Skinfolds were also measured with a skinfold caliper (Lange, Beta Technology, Santa Cruz, CA, USA) at the biceps, triceps, subscapular, suprailiac, supraspinal, lateral and medial calf. According to Frisancho (2008), the total upper-arm and calf area, upper-arm and calf muscle area and upper-arm and calf fat area were calculated [43,44]. All the anthropometric measurements were carried out by the same operator, specifically trained according to a standardized protocol [43,45].

### 2.4. Body Composition 

The impedance measurements were performed with a bioimpedance analyzer (BIA 101 Anniversary, Akern, Florence, Italy) at a frequency of 50 kHz. The accuracy of the BIA instrument was validated before each test session following the manufacturer’s instructions. The participants were assessed in the supine position with legs (45° compared to the median line of the body) and arms (30° from the trunk) abducted. After cleansing the skin with alcohol, two electrodes were placed on the right hand and two on the right foot. Body composition parameters were estimated using specific bioimpedance-derived equations [46,47,48]. 

Fat percentage (%F) = [(4.950/D) − 4.500] × 100Fat mass (FM) = (%F × weight)/100Fat free mass (FFM) = Weight − FM

In addition, bioimpedance values were analyzed according to classic and specific BIVA methods [37,49,50,51]. 

### 2.5. Physical Test 

Right and left handgrip strength was measured with a dynamometer (Takei Scientific Instruments Co., Niigata City, Japan) in a sitting position at a 90-degree flexion of their elbow. Each participant performed three trials with a 1 min rest period between each test. The highest value of all three measurements was used for analysis. To avoid any confounding effect of time of day, all test sessions were performed in the morning, at the baseline and after three months [52]. To assess the strength and the endurance of the lower limbs, the chair stand test has been executed. Before each test, the operator gave orally clear and simple instructions and demonstrated the test. Participants were allowed one practice trial before the actual measurements. A standard chair without armrests was used for all the participants. Participants were instructed to sit in the middle of the chair, back straight, feet approximately shoulder-width apart and placed on the floor at an angle slightly back from the knees with one foot slightly in front of the other to help to maintain balance when standing. Instructions to participants were to stand up and sit down again as many times as possible for 30 s. Participants were encouraged to continue to sit and stand throughout the test. The number of repetitions was recorded, and represented the units for this measure [53]. The Six-Minute Walk Test (6MWT) is a simple test to measure exercise capacity. Participants had to walk for six minutes and they were instructed to go their gait and to slow down or stop if they became fatigued, but to resume once able [54]. A lap was recorded each time the subject passed the starting position. Using an even-toned encouraging phrase, the time remaining in the test was reported to the participants at one-minute intervals. The timer was not stopped if the participants needed to rest. Once the six minutes concluded, the participants were instructed to stop and remain stationary while the end point was marked. Once marked, the total distance walked was calculated in meters [54].

### 2.6. Questionnaire Post the Training Period

At the end of the study, a questionnaire was administered to the participants, in order to understand their habits about general physical activity, their attitude towards green urban space and how they feel after the participation in outdoor training. The questionnaire was validated in a previous study [55]. Attitude represents a synthetic assessment of a psychological object evaluated in positive or negative dimensions [56,57]. The survey was divided into three subsections: (1) physical activity habits and feeling after having done outdoor activities (NW), (2) attitude toward green space components and (3) their evaluation of the park characteristics. The statements were evaluated using the Likert scale, from 1 to 5, in which 1 meant “strongly disagree” and 5 “strongly agree”.

### 2.7. Statistical Analysis

All statistical analyses were performed with Statistica for Windows, versión 8.0 (Stat Soft Italia srl, Vigonza, Padua, Italy). A post hoc analysis was assessed to compute an achieved power given alfa = 0.05 sample size = 77, effect size = 0.43. The test family select was t-test for means difference within groups (matched pairs). The final statistical power was 0.97. Descriptive analysis and independent Student *t*-test were used to assess baseline characteristics and gender differences. Each result was reported as the variable Mean ± Standard Deviation (SD) at two different times (Pre and Post). The Shapiro–Wilk test was used to check the normal distribution of each body composition and physical test variable. When variable data did not distribute as a Gaussian curve, a transformation function (natural logarithm) was applied to reduce the curve skewness. Longitudinal differences were calculated as post-pre among groups for each variable, and mean ± SD, paired Student’s test (*t*) and probability (*p*) values were outlined. Additionally, analysis of variance (ANOVA) was performed to evaluate the differences between male and female groups in the answers to the questionnaire, and the Snedecor–Fisher (F) and probability values (*p*) were reported.

Statistical significance was set at *p* < 0.05.

## 3. Results

### 3.1. Baseline Characteristics of the Participants

Eighty-two people decided to participate in this study, but five people did not complete the period of training so they were excluded from the study. Therefore, 77 participants did the measurements before and after the period of training. The flow chart with a schematic representation of participant allocation is presented in Figure 1.

The largest part of the sample was composed of females (56, 72.7%), and their mean age was 55.53 ± 9.73 years. Men who participated in the study (21, 27.3%) were older than women (60.51 ± 8.15 years vs. 55.53 ± 9.73). Entire sample characteristics are presented in Table 1, and women and men characteristics are presented in Table 2.

### 3.2. Effects of NW on Anthropometric Characteristics

The entire sample and women’s sample showed significant differences in several anthropometric characteristics between the baseline and follow-up (Table 1 and Table 2).

Generally, a decrease in fat parameters was observed: a decrease in the triceps (entire sample: pre = 19.53 ± 0.48, post = 18.36 ± 0.48, women: pre 20.55 ± 3.43, post 19.29 ± 3.85), biceps (entire sample pre 12.09 ± 0.49, post 11.08 ± 0.49, women pre 12.52 ± 4.62, post 11.25 ± 4.62), subscapular (entire sample pre 18.07 ± 0.72, post 15.26 ± 0.54, women pre 17.00 ± 6.54, post 14.55 ± 4.66) and supraspinal (entire sample pre 19.29 ± 0.70, post 17.61 ± 0.72, women pre 18.42 ± 6.07, post 16.89 ± 6.00) skinfolds was shown. Hip circumference (entire sample pre 102.69 ± 0.89, post 101.74 ± 0.91, women pre 101.94 ± 8.69, post 100.71 ± 8.78), calf fat area (entire sample pre 30.83 ± 0.79, post 26.71 ± 1.05, women pre 32.14 ± 6.01, post 27.72 ± 6.01), FM (entire sample pre 26.46 ± 0.77, post 24.96 ± 0.82, women pre 24.48 ± 6.30, post 23.06 ± 7.07) and %F (entire sample pre 37.04 ± 0.38, post 35.57 ± 0.41, women pre 36.85 ± 3.57, post 35.46 ± 3.76) showed a significant decrease, too. On the contrary, arm muscle area showed a significant increase (women pre 38.99 ± 11.35, post 41.78 ± 12.01). In men the variations were more contained, showing a significant decrease only in subscapular skinfold (pre 20.90 ± 4.65, post 17.14 ± 4.52), FM (pre 31.64 ± 4.68, post 30.29 ± 3.89) and %F (pre 37.32 ± 2.84, post 35.90 ± 3.10) after the period of NW training.

### 3.3. Effects of NW on Physical Tests

Regarding the physical test, women and the entire sample presented significant differences in two of them: the 30′′ squat test (entire sample: pre 15.15 ± 0.45, post 17.39 ± 0.44; women: pre 15.58 ± 3.80, post 17.45 ± 3.71) and the six-minute walking test (entire sample: pre 537.16 ± 8.99, post 573.49 ± 7.78; women: pre 540.36 ± 70.52, post 577.53 ± 66.10) showed significant increases after the training period. The handgrip did not show significant improvement after the period of NW training. Men showed a significant increase only in the squat test (pre 14.00 ± 3.57, post 17.52 ± 3.63).

### 3.4. Effects of NW on BIVA

The entire sample and the sample divided by sex presented a significant increase in the phase angle after the period of training (entire sample pre 6.32 ± 0.15, post 7.13 ± 0.20, women pre 6.27 ± 0.94, post 7.06 ± 1.46, men pre 6.29 ± 1.29, post 7.31 ± 1.24). Meanwhile, resistance and reactance showed a change from the baseline, even if the changes were not significant.

### 3.5. Questionnaire

Table 3 shows the participants’ answers.

Only three items (“I prefer to do outdoor physical activity”, “Tax dollars should be spent on nature parks” and “I enjoy talking with neighbours at local nature parks”) showed significant differences. Men had higher scores in the first two items (men 4.92 ± 0.28 and 5.0 ± 0.00, women 4.37 ± 0.94 and 4.48 ± 0.88) and women had higher scores in the item “I enjoy talking with neighbours at the local nature park” (women 3.39 ± 1.23, men 2.54 ± 1.27).

## 4. Discussion

NW is a particular kind of walking in which specially designed poles are used and actively involve the upper body and arms [12]. In recent years, it has gained popularity worldwide as a health-promoting activity [7]. NW is frequently recommended for the elderly with chronic diseases. Despite the evidence, there are few studies on the effects of a period of NW training for the nonclinical middle age and elderly population, for which, considering the efficacy of this activity it is important to have indications.

The main aims of the present study were to evaluate the effectiveness of a period of NW training in a healthy middle-aged population and evaluate the appreciation in doing outdoor PA. Regarding the first goal, men and women showed different effects. Women showed a significant decrease in fat parameters, as skinfolds, calf fat area FM and %F and in hip circumference. In addition, the arm muscle area increased. Men showed a significant decrease only in a skinfold, in FM and %F. Both men and women presented a significant increase in lower body strength, and women showed an increase also in the six-minute walking test.

Regarding the previous literature about the positive effects of NW, several studies showed that NW influenced more the cardiorespiratory fitness than the normal walking, because of the use of poles that involved a higher amount of muscle mass [12]. NW has positive effects on chronic diseases such as diabetes or obesity [11], by benefits the resting heart rate, blood pressure, maximal oxygen consumption, exercise capacity and quality of life [3,13]. Most of the participants to the studies about this kind of physical activity were mid to older aged men and women from the clinical population (diabetes, cardiovascular disease, muscle-skeletal conditions, Parkinson’s disease, etc.) [15,16,17,18,19,20,21,22,23]. Only two studies considered the nonclinical population, but both populations consisted of elderly women [11,24]. Both studies compared the efficacy of a period of NW and walking training, but women in the study by Figarde-Faber et al. (2010) were obese and middle-aged and in the study by Kukkonen-Harjula et al. (2017) were sedentary and aged 50–60 years [11,24]. The results of the two studies are in contrast, because in the first the authors found that within the same walking time, the use of NW poles provided an increase in the intensity and of the energy expenditure. On the contrary, Kukkone-Harjula et al. (2007) showed that NW did not improve upper body muscle strength more than walking and, in addition, they assumed that the poles in NW, used as a support, reduced the training effects on lower extremities [24]. However, in both studies, NW emerges among the safest kind of physical activity [11,24,58]. In fact, Kukkonen-Harjula et al. (2007) found that injuries rate of NW and walking (1.4 NW and 1.9 walking) were lower than that of other kind of activities (volleyball, swimming, tennis, lifestyle activity, etc.) [24]. So, walking and NW were among the safest kinds of activity [58]. Figarde-Fabre et al. (2010) found that the obese women of their study increased their stability thanks to the use of the poles [11]. This is an important feature of NW, that can be practiced with a very low risk of injury and as a primary kind of prevention.

The results of Kukkon-Harjula et al. (2017) are partially in contrast with the results of the present study [24]. In fact, similarly to the study by Kukkon-Harjula et al. (2017) we found no significant differences in the handgrip test, but a significant decrease in the arm’s skinfolds and an increase in the arm’s circumference were observed in our study [24]. Maybe the period of three months is too short to observe significant change in upper body muscle strength, but it may be a sufficient time to observe changes in the body composition of the arm. This is in line with previous studies which showed that NW is a more complete activity than normal walking, due to the use of poles that involve the upper part of the body [14,59,60,61]. In addition, all the participants showed on average an increase in the number of squats done in 30 s, and this could be linked with an increase in the strength of the lower body. This result is in line with other previous studies [61,62]. Regarding the last physical test, women did more meters in the post training six-minute walking test, showing an increase in capacity of resistance. This could be linked to the result of Figarde-Fabre et al. (2010), which found that the subjects perceived the NW as less demanding than the walking without the poles [11]. This could be a positive aspect, because could drive people to continue physical activity for a longer period and to be more active. For this reason, NW could also be considered a primary kind of prevention.

Usually, for middle-aged people, the more common kind of physical activity proposed is walking and resistance training, because they reverse the adverse effects of aging in cellular integrity and function [32,40,63]. Resistance training improve bioimpedance parameters and induce changes in cellular volume and cellular potential [31,32,40,63]. As regards the phase angle, a significant increase was observed both in women and in men, in accordance with the results of resistance training. Several studies have considered the phase angle and its relationship with the health status [38,64]; in particular, the increase in this parameter was associated with an increase in strength and the alteration in cellular membrane integrity or body fluid or a combination of both. So, NW could be considered a “protective” activity for the aging process. In addition, Takeshima et al. (2013) compared the effects of NW, conventional walking and band-based resistance exercise in older adults [62]. They observed that conventional walking only improves cardiorespiratory fitness and the resistance training improve muscle strength, so NW could be a combination of the two kind of activities and provide improvements to each of those components [62]. In addition, performing NW take less time than performing the same amount of conventional walking with additional resistance training sessions. Some studies showed that NW increased also the stability and the dynamic balance, which is an important aspect, especially considering the aging process [16].

Regarding the second goal of the study, the first thing to consider is that the popularity of this sport has increased a lot in the last ten years and mainly in this last two years, may be due to the SARS COVID-19 pandemic situation. To contain the virus diffusion, the Italian Government enforced quarantine and after this period there was increasing research for outdoor activities, considered safer than the indoor [65]. However, NW could be considered an easily accessible and universal kind of activity. In fact, NW, such as walking, is a universal form of physical activity that is appropriate regardless of sex, ethnic group, age, education or income level. It does not require expensive equipment (poles are cheaper than other kinds of sport equipment), special skills or special facilities. In addition, it is a kind of outdoor physical activity and several studies have highlighted a relationship between exposure to the natural environment and better health perception [66]. Experimental research suggested that performing physical activity in nature has additional benefits in comparison with compared to doing it in an indoor environment [57] and, in In addition, exposure to nature could prove restoration from stress and mental fatigue [58]. This is in line with the results of the present study. In fact, participants appreciated the activity and the fact that it was done outdoors, enjoying the contact with nature. In addition, participants showed a good well-being after the practice of NW, with a decrease in anxiety and stress. NW also represents the possibility of enhancement of social interaction and to improve improving social cohesion in the community [59,60,61,62]. This could be an important aspect, especially for women. According to Richardson et al. (2010) one of the motivations reported by women for the non-use of the park is the fear for their safety [67]. This could in part explain the lower preference of the women of this study, to do outdoor physical activity in comparison with men. Men tended to agree more with the use of tax dollars on nature park. This disagrees with a previous study that suggested that women were more sensitive in the importance of neighborhood [68,69,70]. However, the sample of men of the present study was small and make generalized interpretations difficult. Generally, the total score of the attitude towards green urban space was high and in line with a previous study [55].

As a conclusion, considering the results in the present study, NW seemed to be a good, complete and safe kind of activity. The practice of NW has several benefits, not only for clinical populations, and it could be considered as a primary preventive kind of activity. In addition, it is an easy kind of physical activity that everyone could practice and could improve social interaction and cohesion.

The present study has some limitations that should be addressed. Three months of intervention can be considered a relatively short period. Therefore, it is necessary to determinate whether the results would differ over a longer timeframe (six months or a year). Additionally, the intensity of the training was decided a priori, and it is possible that for some participants the intensity was too low. For future research it might be useful to use other methods to monitor the intensity of training, such as VO2max. The training intervention did not consider diet. Participants were asked to not change their diet habits, but for future research it will be an important aspect to consider.

## 5. Conclusions

The aim of the present study was to evaluate the benefits of a period of NW training in the general population. NW could answer two human needs: the importance of the use of green urban space and the possibility to easily perform physical activity. It is an easy kind of physical activity that could be done by anyone and everywhere in the green urban space. Even if it is usually used for a specific kind of population or for rehabilitation in chronic disease, NW also has a lot of potential benefits for the general population. From the results of our study, we can see that NW increases resistance and lower body strength and also influences the changes in body composition. The popularity of this sport is increasing, probably due to the SARS COVID-19 pandemic situation, because outdoor physical activity is recommended. It would be important to encourage its practice not only to older people but also to younger people because it is an easy sport that anyone could practice anywhere they want.

## Figures and Tables

**Figure 1 ijerph-19-07433-f001:**
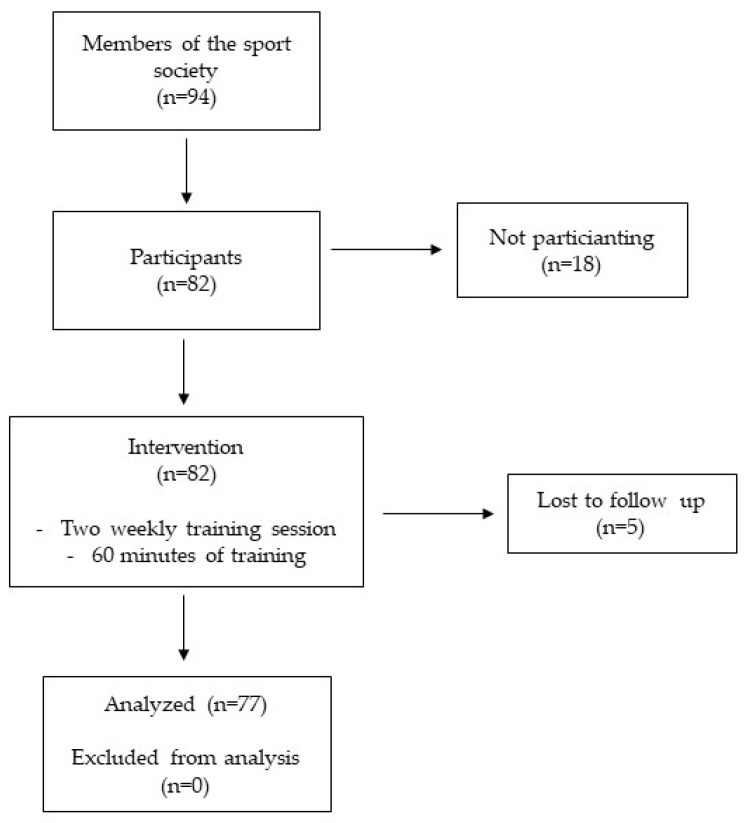
Flow chart.

**Table 1 ijerph-19-07433-t001:** Paired Students’ test and entire sample’s anthropometric measures, body composition and physical test values.

Variables	Pre Mean ± SD	Post Mean ± SD	Differences Post-Pre Mean ± SD	*t*	*p*
Weight, kg	70.81 ± 1.58	70.39 ± 1.57	−0.43 ± 0.63	−0.68	0.50
Height, cm	168.32 ± 8.38	±	±		
BMI, kg/m^2^	24.88 ± 0.44	24.37 ± 0.55	−0.51 ± 0.35	−1.48	0.14
Triceps, mm	19.53 ± 0.48	18.36 ± 0.48	−1.17 ± 0.53	−2.22	**0.03**
Biceps, mm	12.09 ± 0.49	11.08 ± 0.49	−1.01 ± 0.47	−2.16	**0.03**
Subscapular, mm	18.07 ± 0.72	15.26 ± 0.54	−2.81 ± 0.53	−5.29	**0.00**
Suprailiac, mm	20.08 ± 0.71	19.32 ± 0.61	−0.75 ± 0.56	−1.35	0.18
Supraspinal, mm	19.29 ± 0.70	17.61 ± 0.72	−1.67 ± 0.60	−2.77	**0.01**
Medial calf, mm	16.25 ± 0.49	15.52 ± 0.74	−0.73 ± 0.70	−1.05	0.30
Lateral calf, mm	10.45 ± 2.22	10.39 ± 4.84	−0.04 ± 1.31	−0.10	0.65
Arm circumferences, cm	28.83 ± 0.38	29.29 ± 0.40	0.46 ± 0.26	1.80	0.08
Contract arm circumferences, cm	30.16 ± 0.38	30.17 ± 0.38	0.01 ± 0.22	0.02	0.98
Waist circumferences, cm	84.27 ± 1.52	83.03 ± 1.56	1.24 ± 0.80	−1.55	0.13
Hip circumferences, cm	102.69 ± 0.89	101.74 ± 0.91	−0.95 ± 0.40	−2.38	**0.02**
Calf circumferences, cm	37.09 ± 0.32	36.04 ± 0.64	−1.05 ± 0.61	−1.73	0.09
Resistance	513.39 ± 9.50	500.56 ± 9.54	−12.84 ± 13.05	−0.98	0.34
Reactance	55.52 ± 1.04	56.11 ± 1.21	0.59 ± 1.26	0.47	0.64
Phase angle	6.32 ± 0.15	7.13 ± 0.20	0.82 ± 0.19	4.34	**0.00**
FM, kg	26.46 ± 0.77	24.96 ± 0.82	−1.49 ± 0.51	−2.95	**0.00**
FFM, kg	44.28 ± 0.89	44.55 ± 1.09	0.27 ± 0.64	0.42	0.68
%F	37.04 ± 0.38	35.57 ± 0.41	−1.46 ± 0.33	−4.50	**0.00**
Total upper arm area, cm^2^	67.27 ± 1.75	68.61 ± 2.07	1.34 ± 1.51	0.88	0.38
Upper arm muscle area, cm^2^	41.86 ± 1.34	44.57 ± 1.43	2.73 ± 1.05	2.62	**0.01**
Upper arm fat area, cm^2^	25.43 ± 0.76	24.04 ± 0.93	−1.39 ± 0.91	−1.53	0.13
Total calf area, cm^2^	109.73 ± 1.93	107.27 ± 2.50	−2.46 ± 2.06	−1.20	0.24
Muscle area of the calf, cm^2^	78.90 ± 1.62	76.34 ± 2.80	−2.57 ± 2.28	−1.12	0.27
Fat area of the calf, cm^2^	30.83 ± 0.79	26.71 ± 1.05	−4.12 ± 1.09	−3.78	**0.00**
Right handgrip	30.52 ± 1.24	30.59 ± 1.18	0.06 ± 0.67	0.09	0.93
Left handgrip	30.09 ± 1.13	30.01 ± 1.13	−0.08 ± 0.60	−0.14	0.89
6 min Walking test, m	537.16 ± 8.99	573.49 ± 7.78	36.34 ± 8.54	4.25	**0.00**
Squat test, n	15.15 ± 0.45	17.39 ± 0.44	2.24 ± 0.57	3.94	**0.00**

Note. In bold, *p* < 0.05.

**Table 2 ijerph-19-07433-t002:** Paired Students’ test and participants’ anthropometric measures, body composition and physical test values.

	Women					Men				
Variables	Pre Mean ± SD	Post Mean ± SD	Differences Post-Pre Mean ± SD	*t*	*p*	Pre Mean ± SD	Post Mean ± SD	Differences Post-Pre Mean ± SD	*t*	*p*
Weight, kg	65.65 ± 11.69	65.34 ± 12.05	−0.31 ± 0.36	0.40	0.69	84.58 ± 8.99	83.86 ± 7.52	−0.72 ± 1.47	0.70	0.49
Height, cm	165.23 ± 7.07	165.23 ± 7.07	-	-	-	176.54 ± 6.89	176.54 ± 6.89	-	-	-
BMI, kg/m^2^	24.01 ± 3.80	23.39 ± 5.03	−0.62 ± 1.22	1.36	0.18	27.19 ± 3.02	26.97 ± 2.81	−0.17 ± 0.21	0.61	0.55
Triceps, mm	20.55 ± 3.43	19.29 ± 3.85	−1.27 ± 0.42	2.26	0.03	16.81 ± 4.86	15.90 ± 4.31	−0.90 ± 0.55	0.73	0.48
Biceps, mm	12.57 ± 4.62	11.25 ± 4.62	−1.32 ± 0.22	2.40	0.02	10.81 ± 3.09	10.62 ± 4.09	−0.19 ± 1.00	0.21	0.83
Subscapular, mm	17.00 ± 6.54	14.55 ± 4.66	−2.45 ± 1.88	4.04	0.00	20.90 ± 4.65	17.14 ± 4.52	−3.76 ± 0.13	3.48	0.00
Suprailiac, mm	19.05 ± 6.40	18.32 ± 5.34	−0.73 ± 1.06	1.13	0.26	22.81 ± 4.88	22.00 ± 4.51	−0.81 ± 0.38	0.72	0.48
Supraspinal, mm	18.42 ± 6.07	16.89 ± 5.98	−1.54 ± 0.09	2.28	0.03	21.57 ± 5.86	19.52 ± 6.78	−2.05 ± 0.92	1.55	0.14
Medial calf, mm	17.63 ± 3.44	16.58 ± 6.00	−1.04 ± 2.56	1.35	0.18	12.57 ± 4.32	12.68 ± 7.00	0.10 ± 2.68	−0.07	0.95
Lateral calf, mm	12.57 ± 4.32	12.68 ± 7.00	0.10 ± 2.68	−0.07	0.95	8.32 ± 2.68	8.10 ± 0.12	0.22 ± 2.56	0.10	0.45
Arm circumferences, cm	28.38 ± 3.55	28.97 ± 3.45	0.60 ± 0.10	−2.53	0.01	30.10 ± 1.97	30.13 ± 3.48	0.05 ± 1.50	−0.08	0.94
Contract arm circumferences, cm	29.44 ± 3.42	29.54 ± 3.40	0.09 ± 0.02	−0.40	0.69	32.09 ± 2.42	31.87 ± 2.56	−0.22 ± 0.04	0.40	0.69
Waist circumferences, cm	79.66 ± 11.58	78.49 ± 12.17	−1.18 ± 0.59	1.18	0.24	96.56 ± 9.53	95.14 ± 9.42	−1.41 ± 0.11	1.09	0.29
Hip circumferences, cm	101.94 ± 8.69	100.71 ± 8.78	−1.22 ± 0.09	2.43	0.02	104.68 ± 4.52	104.46 ± 4.44	−0.21 ± 0.08	0.39	0.70
Calf circumferences, cm	36.58 ± 2.62	35.62 ± 5.17	−0.95 ± 2.54	1.58	0.12	38.15 ± 3.02	37.10 ± 6.40	−1.05 ± 3.38	0.77	0.45
Resistance	527.61 ± 73.84	510.97 ± 68.28	−16.64 ± 5.56	0.75	0.46	485.20 ± 52.15	473.74 ± 74.65	−11.46 ± 22.49	0.60	0.56
Reactance	57.18 ± 7.15	56.30 ± 9.13	−0.87 ± 1.98	0.65	0.52	52.90 ± 9.76	57.11 ± 8.97	4.21 ± 0.80	−1.55	0.15
Phase angle	6.27 ± 0.94	7.06 ± 1.46	0.79 ± 0.52	−3.49	0.00	6.29 ± 1.29	7.31 ± 1.24	1.02 ± 0.06	−2.58	0.03
FM, kg	24.48 ± 6.30	23.06 ± 7.07	−1.43 ± 0.78	2.19	0.03	31.64 ± 4.68	30.29 ± 3.89	−1.35 ± 0.78	2.56	0.02
FFM, kg	41.17 ± 5.82	41.16 ± 8.33	−0.01 ± 2.51	0.01	0.99	52.94 ± 5.29	54.06 ± 5.21	1.18 ± 0.09	−1.05	0.31
%F	36.85 ± 3.57	35.46 ± 3.76	−1.39 ± 0.19	3.92	0.00	37.32 ± 2.84	35.90 ± 3.10	−1.42 ± 0.26	2.22	0.04
Total upper arm area, cm^2^	65.10 ± 16.67	66.55 ± 18.55	1.46 ± 1.89	1.45	0.40	72.32 ± 9.21	74.36 ± 15.59	2.04 ± 6.38	−0.31	0.76
Upper arm muscle area, cm^2^	38.99 ± 11.35	41.78 ± 12.01	2.80 ± 0.66	2.79	0.02	49.36 ± 8.81	52.39 ± 10.30	3.03 ± 1.49	−1.05	0.31
Upper arm fat area, cm^2^	26.11 ± 6.72	24.77 ± 7.95	−1.34 ± 1.22	1.37	0.18	22.96 ± 6.47	21.97 ± 8.36	−0.99 ± 1.89	0.71	0.49
Total calf area, cm^2^	107.06 ± 15.48	105.18 ± 18.13	−1.88 ± 2.65	1.06	0.29	116.55 ± 18.43	113.14 ± 29.64	−3.42 ± 11.21	0.66	0.51
Muscle area of the calf, cm^2^	74.92 ± 11.76	71.72 ± 21.91	−3.19 ± 10.15	1.27	0.21	90.07 ± 14.19	89.26 ± 26.80	−0.81 ± 12.60	0.16	0.88
Fat area of the calf, cm^2^	32.14 ± 6.01	27.72 ± 6.01	−4.42 ± 3.49	3.48	0.00	26.49 ± 8.22	23.88 ± 7.69	−2.61 ± 0.53	1.51	0.15
Right handgrip	25.89 ± 6.00	26.05 ± 5.80	0.17 ± 0.19	−0.95	0.35	42.83 ± 10.34	42.05 ± 10.36	−0.78 ± 0.62	0.59	0.57
Left handgrip	25.41 ± 4.96	25.44 ± 5.70	0.03 ± 0.75	−0.87	0..39	41.90 ± 8.54	41.29 ± 8.61	−0.62 ± 0.07	0.68	0.51
6 min Walking test, m	540.36 ± 70.52	577.53 ± 66.10	37.16 ± 4.42	4.19	0.00	531.98 ± 95.20	564.02 ± 64.45	32.04 ± 27.75	−1.53	0.14
Squat test, n	15.58 ± 3.80	17.45 ± 3.71	1.87 ± 0.09	−2.74	0.00	14.00 ± 3.57	17.52 ± 3.63	3.52 ± 0.06	−3.06	0.00

Note: SD = standard deviation, *p* = *p*-value.

**Table 3 ijerph-19-07433-t003:** Participants’ answers to the questionnaire and differences between sexes (ANOVA).

Variables	Female Mean (SD)	Male Mean (SD)	F	*p*
I prefer to do outdoor physical activity	4.37 (0.94)	4.92 (0.28)	4.38	0.04
Green space in cities is important	4.52 (0.86)	4.92 (0.28)	2.76	0.10
Nature parks improve quality of life	4.50 (0.86)	4.62 (0.65)	0.20	0.65
Tax dollars should be spent on nature parks	4.48 (0.88)	5.00 (0.00)	4.41	0.04
Contact with nature is important for well-being	4.50 (0.86)	4.85 (0.38)	1.98	0.16
It is important to have convenient nature parks in cities	4.52 (0.86)	4.77 (0.60)	0.98	0.32
Nature parks in the cities provide valuable contacts with nature	4.31 (0.99)	4.15 (0.80)	0.30	0.59
The time spent in an urban nature park relaxes you	4.35 (0.97)	4.69 (0.63)	1.43	0.24
I expect to feel refreshed after visiting a nature park	4.24 (1.03)	4.23 (0.93)	0.00	0.97
I enjoy talking with neighbours at local nature park	3.39 (1.23)	2.54 (1.27)	4.92	0.03
I like the structure of the park you use	3.59 (1.06)	3.46 (0.88)	0.17	0.68
After outdoor physical activity you feel.				
Physical well-being	4.54 (0.99)	4.69 (0.48)	0.29	0.59
Psychological well-being	4.54 (0.97)	4.54 (0.66)	0.01	1.00
Lessening of anxiety	4.41 (1.00)	4.38 (0.87)	0.02	0.95
Lessening of stress	4.43 (1.00)	4.46 (0.66)	0.02	0.90
Personal satisfaction (physical)	4.44 (1.06)	4.62 (0.51)	0.32	0.57
Mood improvement	4.52 (0.99)	4.38 (0.77)	0.20	0.65
Fatigue	2.67 (1.29)	3.46 (1.27)	3.89	0.05
General well-being	4.56 (0.96)	4.85 (0.38)	1.08	0.30
How do you consider your physical health?	3.67 (0.89)	3.77 (0.60)	0.27	0.61
How do you consider your mental health?	3.89 (0.77)	3.77 (0.87)	0.10	0.75

Note: SD = standard deviation, *p* = *p*-value.

## Data Availability

The data presented in this study are available on request from the first author.

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
