# Peer review of "Effects of Nordic Walking Training on Anthropometric, Body Composition and Functional Parameters in the Middle-Aged Population"

_ijerph, 2022, doi:10.3390/ijerph19127433_

Round 1

Reviewer 1 Report

Comments for the author

IJERPH-1766307 v1

Title: Effects of Nordic Walking Training on Anthropometric, Body Composition and Functional Parameters in the Middle-aged Population

Article Type: Original Article

Keywords:

 Anthropometric measure; body composition; middle-aged; Nordic Walking; outdoor physical exercise; physical exercise

  GENERAL:

This study tries to reveal the effects of a 3 months period Nordic Walking (NW) training in the non-clinical middle-aged and elderly population on anthropometric, body composition and functional parameters. Although many studies have been conducted examining the effects of varied training interventions on middle-aged and elderly populations, this article could contribute in our knowledge especially about healthy middle-aged and elderly populations.

The study itself has been accomplished in a suitable manner, and there is little to comment on the introduction, methods, results or discussion of the findings.

There are some minor comments and suggestions, which are detailed below.

Many phrases in the document  may be wordy. Consider changing the wording.

There are many missuses of commas. The passive voice isn’t an error, but it may be less clear and compelling in academic writing. Consider rewriting in the active voice.  

SPECIFIC:     Introduction Line 36: The phrase “Worldwide as a result.” may be wordy. Consider changing the wording with “due to”. Line 37: The phrase “global risks for mortality in the world” may be wordy. Consider changing the wording with “global risks for mortality globally”. Line 40: The phrase “in order to” may be wordy. Consider changing the wording with “to”. Line 41: It appears that " population” may be unnecessary in this sentence.  Consider removing it. Line 41-42: The phrase “in order to” may be wordy. Consider changing the wording with “to”. Line 42: It appears that " the” may be unnecessary in this sentence.  Consider removing it. Line 60: It appears that the phrase " In fact,” may be unnecessary in this sentence.  Consider removing it. Line 81: The sentence “The vector slope represented by PhA is an indicator of...” may be wordy. Consider rephrasing to “The vector slope indicates…”     Materials & Methods Line 99:decide” seems to be in the wrong tense Consider correct it to “decided Line 105: it seems that there is an article usage problem here . Consider adding the article “the University…”. Line 116: It appears that " both” may be unnecessary in this sentence.  Consider removing it.   Line 120: The plural verb “were” does not appear to agree with the singular subject “Cut-off”. Consider changing the verb form for subject -verb agreement. Line 128: It appears that you type “are”. Consider correct it to “area Line 150: The phrase “In order to” may be wordy. Consider changing the wording with “To”. Line 153: The phrase “Prior to” may be wordy. Consider changing the wording with “Before”. Line 162: The phrase “and they represented” may be wordy. Consider changing the wording with “representing”.   Results   Line 208: It appears that you type “anthropmetric”. Consider correct it to “anthropometric Line 225: It appears that you type “increasing”. Consider correct it to “increase Line 239: It appears that you type “significative”. Consider correct it to “significant   Discussion   Line 245: It appears that " that” may be unnecessary in this sentence.  Consider removing it.   Line 246: it seems that there is an article usage problem here “…elderly”. Consider adding the article “… for the elderly…”. Line 251: it seems that there is an article usage problem here “…healthy”. Consider adding the article “… a healthy…”. Line 252: It appears that the phrase " In fact,” may be unnecessary in this sentence.  Consider removing it. Line 299: It appears that the phrase "also be” may be rephrased to “be also”.  Consider rephrasing it.   Line 303: The plural verb “induce” does not appear to agree with the singular subject “resistance training”. Consider changing the verb form for subject -verb agreement. Line 306: The phrase “taken into account” may be wordy. Consider changing the wording with “considered”. Line 309: It appears that the phrase " kind of” may be unnecessary in this sentence.  Consider removing it. Line 311: It appears that you type “adult”. Consider correct it to “adults Line 312: It appears that you type “improve”. Consider correct it to “improves Line 313: It appears that you type “kind”. Consider correct it to “kinds   Line 316: It appears that the phrase "increased also ” may be rephrased to “increased also”.  Consider rephrasing it.   Line 317: it seems that there is a pronoun usage problem here “that is”. Consider rephrasing to “which is”. Line 326: It appears that you type “other kind”. Consider correct it to “other kinds Line 326: It appears that you type “skill”. Consider correct it to “skills Line 329: The sentence “Experimental research suggested that performing physical activity in nature have additional benefits in comparison with doing it in an indoor environment [57] and, in addition, exposure to nature could prove restoration from stress and mental fatigue [58].” may be unclear or hard to follow. Consider rephrasing to “Experimental research suggested that performing physical activity in nature haves additional benefits in comparison with compared to doing it in an indoor environment [57] and, in In addition, exposure to nature could prove restoration from stress and mental fatigue [58].   Line 335-336: The sentence “NW represents also the possibility of enhancement of social interaction and to improve social cohesion in the community [59–62].” may be unclear or hard to follow. Consider rephrasing to “NW also represents the possibility of enhancementing of social interaction and to improve improving social cohesion in the community [59–62].” Line 339: It appears that you type “of this study”. Consider correct it to “in this study   Line 347-349: The sentence “The practice of NW has several benefits not only for the clinical population and it could be considered as a primary preventive kind of activity.” may be unclear or hard to follow. Consider rephrasing to “The practice of NW has several benefits not only for clinical populations and it could be considered as a primary preventive kind of activity.”   Conclusions

Line 363: It appears that the phrase " This section is mandatory” may be unnecessary in this sentence.  Consider removing it.

Reviewer 2 Report

The manuscript entitled "Effects of Nordic Walking Training on Anthropometric, Body Composition and Functional Parameters in the Middle-aged Population" shows that a 3-months of Nordic Walking (NW) is effective for reducing body fat, anxiety, and stress in healthy middle-aged individuals. This before-after study may provide useful information on the effectiveness of NW for maintaining metabolic and psychological health to health care providers; however, there are several concerns to be addressed for publication in the journal.

1.      The authors stated that the improvement in body fat parameters was more significant in women than men; however, the difference in sample size between women (n = 56) and men (n = 21) could affect this result. The reviewer would suggest that the authors analyze the data of the entire cohort in addition to the sex-separated analysis.

2.      Also, the authors should perform sample size calculation to assess whether this study has a sufficient statistical power to detect the effect of NW on body composition and functional parameters.

3.      What was the sampling method used in this study?

4.      The authors did not mention the intensity of NW training. Please clarify it using appropriate indices such as VO2max, heart rate reserve, or rate of perceived exertion.

5.      The validity and accuracy of the bioimpedance analysis device should be described with specific numbers (e.g., correlation to the reference method, sensitivity, specificity, or ROC analysis). The authors are required to cite academic paper(s) of the validation study.

6.      The authors should describe diet (energy intake) habit of study participants during the study period because diet has a substantial impact on body composition. How did the authors adjust energy intake during the study period?

7.      The authors performed a questionnaire survey regarding general physical activity at the end of the study. In my opinion, this survey should be performed at baseline to avoid biases due to the intervention. Please explain.

8.      In the Introduction section, the authors described, “Previous studies have shown that the PhA can be modulated by exercise.” How does exercise modulate phase angle? Please explain in more detail.

9.      The authors need to mention the limitations of this study in the Discussion (e.g, study design, methodology, confounding factors and so on).

Round 2

Reviewer 2 Report

The authors have adequately responded to my comments.